

# Charge order from structured coupling in VSe$_2$

Jans Henke[1], Felix Flicker[2,3], Jude Laverock[4] and Jasper van Wezel[1*]

**1** Institute for Theoretical Physics Amsterdam and Delta Institute for Theoretical Physics,
University of Amsterdam, Science Park 904, 1098 XH Amsterdam, The Netherlands
**2** School of Physics and Astronomy, Cardiff University, Cardiff CF24 3AA, United Kingdom
**3** School of Mathematics, University of Bristol, Bristol BS8 1TW, United Kingdom
**4** H. H. Wills Physics Laboratory, University of Bristol,
Tyndall Avenue, Bristol BS8 1TL, United Kingdom

⋆ vanwezel@uva.nl

## Abstract

Charge order–ubiquitous among correlated materials–is customarily described purely as an instability of the electronic structure. However, the resulting theoretical predictions often do not match high-resolution experimental data. A pertinent case is 1$T$-VSe$_2$, whose single-band Fermi surface and weak-coupling nature make it qualitatively similar to the Peierls model underlying the traditional approach. Despite this, its Fermi surface is poorly nested, the thermal evolution of its charge density wave (CDW) ordering vectors displays an unexpected jump, and the CDW gap itself evades detection in direct probes of the electronic structure. We demonstrate that the thermal variation of the CDW vectors is naturally reproduced by the electronic susceptibility when incorporating a structured, momentum-dependent electron-phonon coupling, while the evasive CDW gap presents itself as a localized suppression of spectral weight centered above the Fermi level. Our results showcase the general utility of incorporating a structured coupling in the description of charge ordered materials, including those that appear unconventional.


# 1   Introduction

Density waves of charge, spin, or orbital occupation play a central role in determining the physical properties of many materials, ranging from elements [1,2], to cuprate high-$T_c$ superconductors [3–5], pnictides [6,7], complex oxides [8–10], and (multi)ferroics [11–14]. Understanding the mechanisms driving density wave formation is important for understanding their interplay with other types of order, and is key in tuning phases of matter to obtain ideal properties for applications [15–17]. Here, we focus on the most common mechanism underlying charge density wave (CDW) order, based on the combination of an electronic instability and atomic distortions cooperatively driving the CDW formation, in qualitative analogy to the idealized Peierls model [18]. It was recently argued that a broadly applicable theoretical framework taking into account the structured, momentum-dependent electron-phonon coupling may supplement the traditional analysis based solely on nesting of the electronic structure, and yield *quantitative* agreement with experimental observations [19,20]. Going beyond the strongly-coupled settings considered before, we show here that this approach resolves several paradoxes surrounding the CDW phase in the weakly-coupled compound 1$T$-VSe$_2$.

The CDW gap in VSe$_2$ was found in recent scanning tunnelling spectroscopy (STS) measurements to be $2\Delta = 24{\pm}6$ meV at a temperature of 5 K [17], while the critical temperature is approximately $T_c \approx 110$ K [15,21–23]. This is close to the BCS ratio of $2\Delta(T{=}0) = 3.52\, k_B T_c$ [24], which together with the lack of evidence for charge-order fluctuations above $T_c$ places VSe$_2$ firmly within the weak-coupling regime [17]. Electronically, a single band of predominantly single-orbital character makes up the Fermi surface (FS), consistent with a model Peierls description. Despite this apparent best-case scenario for a weak-coupling CDW, several experimental observations appear to be paradoxical and inconsistent with the customary interpretation of nesting-driven charge order.

Angle-resolved photo-emission spectroscopy (ARPES) studies, for example, do not show clear gaps in the spectral function at low temperatures, such that the CDW gap structure remains unclear [25–29]. This is in stark contrast to other transition metal dichalcogenides (TMDC) with CDW instabilities, such as 2$H$-NbSe$_2$, 2$H$-TaSe$_2$ and 1$T$-TaS$_2$ [30,31]. Furthermore, while the in-plane components of the three simultaneous CDW wave-vectors $\mathbf{Q}_i$ ($i = 1, 2, 3$) in VSe$_2$ are commensurate (periodicity $4a$ with lattice parameter $a$), it was determined via X-ray diffraction that their common out-of-plane component is incommensurate and varies from $q_z = 0.314\, c^*$ at 105 K to $q_z = 0.307\, c^*$ below 85 K with $c^*$ the reciprocal lattice vector along $k_z$ [21].

The thermal evolution of the ordering wave-vector was deemed anomalous, and led to the suggestion that this material hosts two distinct CDW phases [21,32]. This is unusual, since both phases remain incommensurate, and the transition between them is not of the common lock-in type. Phase contrast in satellite dark field images led to the suggestion that the transition may be between a high-$T$, three-component CDW and a low-$T$ phase with only two of the three symmetry-related $\mathbf{Q}_i$, a so-called 2Q phase [32]. Although theoretically allowed, such a 2Q phase would be unusual, as it can only be stable in a small region of phase space and requires fine-tuned contributions from sixth order terms in a Landau expansion of the

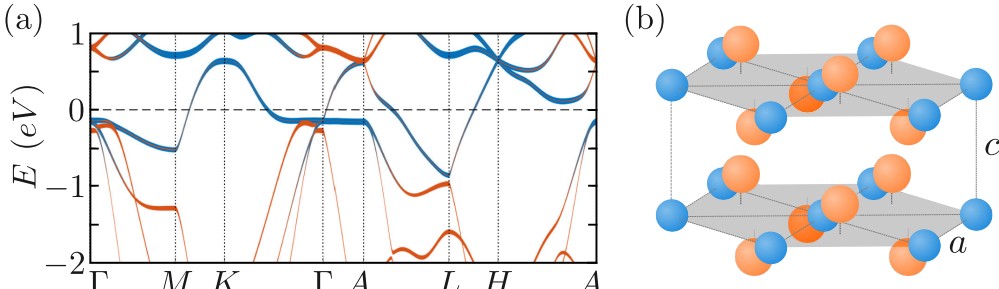

Figure 1: (a) The *ab initio* bandstructure of VSe$_2$ with the orbital character of the bands indicated. Orange corresponds to Se character, and blue indicates V character. The single band crossing the Fermi level was used to compute the electronic susceptibility. (b) Sketch of the layered atomic structure of 1$T$-VSe$_2$.

free energy [20, 32]. Various scanning tunnelling microscopy (STM) experiments (down to 4.2 K) report a 3Q CDW phase [15, 17, 33, 34], while others observe an enhanced intensity of one or two CDW wave-vectors [35, 36]. The latter could be the effect of an anisotropic STM tip [17] and/or spatial variations in the relative phases of the three CDW components [34]. Concrete evidence for a phase transition around 85 K is also absent in thermodynamic probes [15, 22, 23].

## 2 Structured electronic susceptibility

To determine the nature of the CDW instability in VSe$_2$, we first compute its electronic structure using an *ab-initio* calculation within the local density approximation (LDA), based on the all-electron full-potential linear augmented plane wave (FLAPW) Elk code [37]. We used the experimental lattice parameters of $a = 3.356$ Å, $c = 6.104$ Å, and the relative distance of the Se planes from the V planes $z_{Se} = 0.25$ [38]. Relaxation of the Se position did not significantly affect the band structure or the computed value of the Fermi energy ($E = 0$). We used a mesh of $32 \times 32 \times 24$ **k**-points in the full Brillouin zone ($> 2000$ in the irreducible wedge) to achieve convergence. In agreement with earlier computations, our *ab initio* calculations show only a single band crossing the Fermi level, of predominantly Vanadium, 3$d$-orbital character, which significantly disperses along $k_z$ (see Fig. 1). We evaluated the energies for this band on a $100 \times 100 \times 400$ **k**-point mesh for subsequent calculations of the Lindhard response function, as well as the structured susceptibility, which includes a momentum-dependent EPC. Because *ab initio* predictions of the Fermi energy may vary slightly from experimentally observed levels, and because non-stoichiometry and self-intercalation in VSe$_2$ samples are known to affect the precise value of the Fermi energy [39], we shift all of our obtained DFT energies up by 20 meV to obtain a best-fit value for $E_F$ compared to the experimental Fermi level (see also Supplemental Material).

For the EPC matrix elements, we use the expression derived by Varma *et al.*, which has been well-tested for transition metal compounds with $d$-orbital character at $E_F$ [20, 40]. In the case of a single band crossing $E_F$, the expression simplifies to:

$$\mathbf{g}_{\mathbf{k},\mathbf{k+q}} \propto \frac{\partial \xi_{\mathbf{k}}}{\partial \mathbf{k}} - \frac{\partial \xi_{\mathbf{k+q}}}{\partial \mathbf{k}}. \tag{1}$$

Here, $\xi_{\mathbf{k}}$ is the electronic dispersion taken from the density-functional theory calculation. The direction of the vector $\mathbf{g}_{\mathbf{k},\mathbf{k+q}}$ indicates the polarisation of the phonons coupled to. The displacements of Vanadium atoms associated with the CDW transition are known to be purely

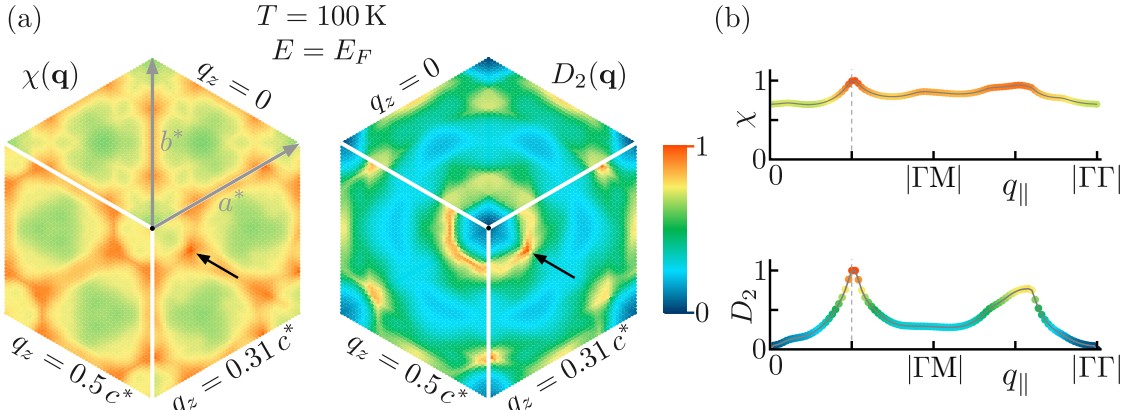

Figure 2: (a) The Lindhard function $\chi(\mathbf{q})$ (left) and the structured susceptibility $D_2(\mathbf{q})$ (right) at $T = 100\,\text{K}$ as a function of in-plane $q_{\parallel}$ (with $q_{\parallel} = 0$ indicated by the black dot), at various values of $q_z$. Both graphs are normalised to their respective maxima, which lie around $\mathbf{q} = (0, 0.25, 0.31)$ rlu and symmetry-related positions (black arrows). (b) Line cuts of the Lindhard function and structured susceptibility at $q_z = 0.31c^*$, varying $q_{\parallel}$ along a line through the black arrows in (a).

in-plane and longitudinal [41]. From here on, we therefore consider only the component of the EPC vector parallel to the in-plane phonon momentum: $g_{\mathbf{k},\mathbf{k+q}} = \mathbf{g}_{\mathbf{k},\mathbf{k+q}} \cdot \mathbf{q}_{\parallel}/|\mathbf{q}_{\parallel}|$.

The structured electronic susceptibility can be derived from a perturbative expansion of the phonon propagator. Since VSe$_2$ falls in the weak-coupling regime, it is sufficient to consider uncorrelated virtual electron-hole excitations. That is, we use the random phase approximation (RPA), and neglect vertex corrections, which should be small [42]. The renormalised phonon propagator is then described by $D_{RPA} = (D_0^{-1} - D_2)^{-1}$, with bare phonon propagator $D_0$ and structured electronic susceptibility $D_2$, given by [19, 43]:

$$D_2(\mathbf{q}) = -\sum_{\mathbf{k}\in BZ} |g_{\mathbf{k},\mathbf{k+q}}|^2 \frac{f(\xi_{\mathbf{k}}) - f(\xi_{\mathbf{k+q}})}{\xi_{\mathbf{k}} - \xi_{\mathbf{k+q}} + i\delta}. \tag{2}$$

Here, $f(\xi)$ is the Fermi-Dirac distribution function and we use a small regulator $\delta = 0.1\,\text{meV}$. The Lindhard function $\chi(\mathbf{q})$ is defined as the above, but taking $g_{\mathbf{k},\mathbf{k+q}} = 1$:

$$\chi(\mathbf{q}) = -\sum_{\mathbf{k}\in BZ} \frac{f(\xi_{\mathbf{k}}) - f(\xi_{\mathbf{k+q}})}{\xi_{\mathbf{k}} - \xi_{\mathbf{k+q}} + i\delta}. \tag{3}$$

The full $g_{\mathbf{k},\mathbf{k+q}}$ enters the renormalised phonon dispersion via $D_2$ in the RPA calculation: $\Omega_{RPA}^2(\mathbf{q}) = \Omega_0^2(\mathbf{q}) - \Omega_0(\mathbf{q})D_2(\mathbf{q})$. Here, $\Omega_0(\mathbf{q})$ is the bare (high-temperature) phonon dispersion [20]. At $T_c$, phonons will exhibit a Kohn anomaly such that $\Omega_{RPA}(\mathbf{Q}_i) = 0$. As long as $\Omega_0(\mathbf{q} \approx \mathbf{Q}_i)$ has no sharp features, the maximum of $D_2(\mathbf{q})$ close to $T_c$ will lie at $\mathbf{q} = \mathbf{Q}_i$.

## 3 The CDW propagation vector

In Fig. 2 we show $\chi(\mathbf{q})$ and $D_2(\mathbf{q})$ for three values of $q_z$, while $q_x$ and $q_y$ span one reciprocal lattice vector each. $D_2(\mathbf{q})$ is not periodic across Brillouin zones, because of the projection of $\mathbf{g}_{\mathbf{k},\mathbf{k+q}}$ onto the in-plane radial direction of $\mathbf{q}$. It is clear from Fig. 2 that $\chi$ disperses significantly less than $D_2$, and is far-removed from a divergence. This is indicative of the small degree of

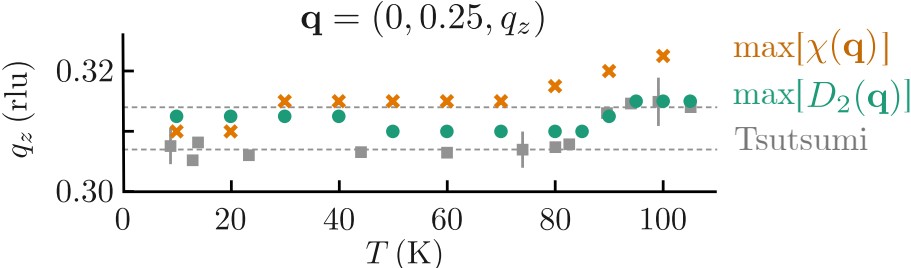

Figure 3: The temperature dependence of the position of the maximum of $\chi$ (crosses) and $D_2$ (circles) along the line $\mathbf{q} = (0, 0.25, q_z)$, plotted alongside experimental data reproduced from Ref. [21] (squares and dashed line fits). Note that $\chi$ is a much flatter function than $D_2$ and that its shape varies strongly with minute changes in the chemical potential (see also Fig. 6 of the Supplemental Figures).

nesting in VSe$_2$ (see also Section A.1). We find that the maxima of both $\chi$ and $D_2$ at $q_z = 0.31$ lie close to $(q_x, q_y) = (0, 0.25)$ (in rlu), in agreement with the experimentally observed in-plane value [21]. Deviations of the peak position of the order of the in-plane $k$-mesh resolution of 0.01 rlu are not expected to have observable consequences, because as long as the peak is close to commensurate, the coupling between the CDW order parameter and the lattice (neglected here) is prone to locking the in-plane component of the CDW wave-vector into the lattice-preferred commensurate value [44, 45]. We expect no lock-in effect in the out-of-plane direction, because the peak in susceptibility is further from low-period commensurate values, inter-layer coupling is weak [17], and the atomic displacements are purely in-plane [41]. This agrees with the observed values of $Q_z$ remaining incommensurate at all temperatures [21].

To assess the out-of-plane position of the peak in the susceptibility, and compare it to the experimentally determined values for the CDW wave-vector, we compute $D_2$ (and $\chi$) along the line $\mathbf{q} = (0, 0.25, q_z)$. Tracking the maximum along this line as a function of temperature yields Fig. 3.[1] We find remarkable agreement between the thermal evolution of the peak position of $D_2$ from 50-105 K and the $q_z$ values observed by Tsutsumi [21]. Importantly, it shows a smooth variation with temperature that quantitatively fits the experimental data points without requiring any discontinuous phase transition. This trend is stable under variation of the chemical potential (shown in Fig. 7). In contrast, although the Lindhard function $\chi$ in Fig. 3 shows a temperature variation similar to $D_2$ for this specific value of $E_F$, even minute changes in the chemical potential yield a qualitatively different thermal evolution of $\chi$.

## 4 The CDW gap

Having established the smooth thermal evolution of the CDW propagation vector, we next turn to its CDW gap structure. Two practical issues contributing to its elusiveness are the small gap size ($2\Delta \approx 24$ meV [17]) and the three-dimensional nature of the electron dispersion, which necessitate experiments with high energy and $k_z$-resolution. Some low-temperature ARPES measurements are suggestive of a gap around $k_z \approx 0.5c^*$ [25, 27–29]. The reported gap size of 80-100 meV in Ref. [25], however, refers to a shift in peak positions of energy dispersion curves, while the CDW gap is more closely related to the leading edge shift [46]. Ref. [27–29], on the other hand, show spectral weight suppression in Fermi pockets around the L-point even above $T_c$, while Ref. [26] has a lower resolution and reports no gaps. The location and shape

---

[1]This is equivalent to finding the CDW ordering wave-vector of VSe$_2$ if we were to quench the system from its high-temperature state directly to the chosen temperature.

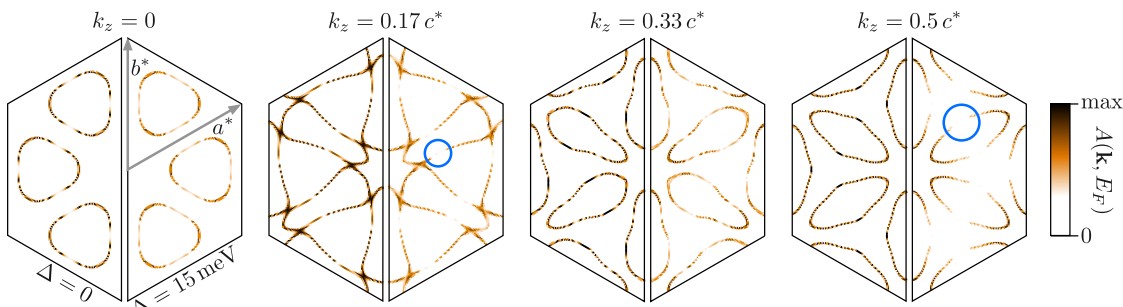

Figure 4: The spectral function of VSe$_2$ at 20 meV above $E_F$, where the effect of the CDW gap is most pronounced. The four plots correspond to different values of $k_z$. In each, the left(right) side shows the ungapped(gapped) phase above(below) $T_c$. All plots employ the same colour scale. We employed a spectral broadening of $\Sigma = 15i$ meV, and a constant gap $\Delta = 15$ meV. The blue circles highlight two gapped regions separated by $\mathbf{Q} = (0, 0.25, 0.33)$.

of the partial CDW gaps in VSe$_2$ thus remain to be determined conclusively.

Based on the energy dispersion of the band that makes up the FS, we can compute the spectral function $A(\mathbf{k}, \epsilon)$ probed by photoemission experiments in a numerically inexpensive way [20]. The electron propagator in the normal state is $G_0(\mathbf{k}, i\omega_n) = (i\omega_n - \xi_\mathbf{k} - \Sigma_\mathbf{k} - \mu)^{-1}$. Here, $i\omega_n$ are Matsubara frequencies, while $\mu$ is the chemical potential. The complex-valued electron self-energy is given by $\Sigma_\mathbf{k} = \Sigma'_\mathbf{k} + i\Sigma''_\mathbf{k}$, and has a real part that shifts the energy of the state at $\mathbf{k}$, while the imaginary part broadens its linewidth. Wick rotating $i\omega_n$ to an energy $\epsilon$ and an infinitesimal imaginary part $i\delta$, we obtain the spectral function, $A_0(\mathbf{k}, \epsilon) = -\frac{1}{\pi}\mathrm{Im}[G_0(\mathbf{k}, \epsilon + i\delta)]$. We assume $\Sigma'_\mathbf{k} = 0$ and for $\Sigma''_\mathbf{k}$, which describes the experimental resolution, we use a constant value of 15 meV. We obtain the normal-state spectral function at $E_F$ ($\epsilon = 0$) shown in the left halves of the hexagonal plots in Fig. 4.

To predict where gaps will open up in the spectral function as the CDW order sets in, we use a similar method to that developed in the context of superconductivity by Nambu [47] and Gor'kov [48]. Rather than constructing a new field theory starting from propagators with the symmetries of the ordered state, we complement the disordered propagators with additional, anomalous electron propagators $F^{\mathbf{k}_1}_{\mathbf{k}_2} = \langle \psi_{\mathbf{k}_1} \psi^\dagger_{\mathbf{k}_2} \rangle$ that do not conserve momentum up to one CDW wave-vector. Here $\psi^\dagger_\mathbf{k}$ creates an electron with momentum $\mathbf{k}$, and $\mathbf{k}_1 - \mathbf{k}_2 = \pm\mathbf{Q}_i$, with $\mathbf{Q}_{1,2,3}$ the three CDW propagation vectors. Since the CDWs in VSe$_2$ are incommensurate, an infinite number of different $F^{\mathbf{k}_1}_{\mathbf{k}_2}$ could be constructed. For the sake of computation, we approximate $Q^z_i = \frac{1}{3}\mathbf{c}^*$ for all propagation vectors. Further noting that $12\mathbf{Q}_i = \mathbf{0}$ and $\sum_i \mathbf{Q}_i = \mathbf{c}^*$, we can represent all possible $\mathbf{k}_1$ and $\mathbf{k}_2$ by $\mathbf{k} + m\mathbf{Q}_1 + n\mathbf{Q}_2$ (with $m, n \in [0, 11]$). Doing so, we generate a matrix $\hat{G}$ whose elements are the renormalised electron propagators $G(\mathbf{k} + m\mathbf{Q}_1 + n\mathbf{Q}_2)$ on the diagonal, and anomalous propagators $F^{\mathbf{k}_1}_{\mathbf{k}_2}$ and $(F^{\mathbf{k}_1}_{\mathbf{k}_2})^\dagger$ for any off-diagonal element with $\mathbf{k}_1$ and $\mathbf{k}_2$ differing by exactly one $\mathbf{Q}_i$. We then construct a matrix Dyson equation:

$$\hat{G} = \hat{G}_0 + \hat{G}_0 \hat{\Sigma} \hat{G} \quad \Rightarrow \quad \hat{G} = (\hat{I} - \hat{G}_0 \hat{\Sigma})^{-1} \hat{G}_0. \tag{4}$$

Here, $\hat{G}_0$ is a diagonal matrix of bare propagators evaluated at momenta $\mathbf{k} + m\mathbf{Q}_1 + n\mathbf{Q}_2$; $\hat{\Sigma}$ is a matrix with self-energies on the diagonal and gaps $\Delta$ in off-diagonal elements connected by one $\mathbf{Q}_i$; and $\hat{I}$ is the identity matrix. Solving this equation for the top left element of $\hat{G}$, we find $G(\mathbf{k})$, and hence the spectral function in the presence of a CDW gap. This method provides an inexpensive way to predict the gap structure of any CDW system, given only the band structure and the ordering wave-vectors.

We find that some portions of the spectral function are suppressed from around approximately 20 meV above $E_F$, as shown in Fig. 4. The offset of this CDW gap from $E_F$ is not surprising, since the structuring of the EPC will generically stabilise a CDW with wave vectors that do not nest the Fermi surface, and therefore do not necessitate any particle-hole symmetry (in contrast, for example, to a superconducting gap). Indeed, asymmetric CDW gaps have been observed in various TMDCs before [49]. The positive offset found here is consistent with STS results, which show a suppression of the density of states with width $2\Delta = (24 \pm 6)$ meV centered at 10 meV above $E_F$ [17].

The clearest gaps in the computed spectral function appear on the long sides of the oval-shaped lobes around the L point ($k_z = 0.5c^*$). These gaps are connected to others at $k_z = 0.17c^*$ by a CDW wave-vector, as indicated by blue circles in Fig. 4. Planes at other $k_z$ values are either unaffected by the CDW or experience a moderate loss of spectral weight. It is probable that the true gap function $\Delta(\mathbf{k})$ in VSe$_2$ depends on $\mathbf{k}$. Obtaining a self-consistent solution for the gap function, which explicitly incorporates the EPC structure, is possible in principle [20], but the three-dimensional nature of the electron dispersion implies a significant computational cost. Moreover, including momentum dependence in $\Delta(\mathbf{k})$ can only change the relative sizes of gaps and will not allow additional gaps to open on top of those already observed in Fig. 4. The locations highlighted by blue circles in Fig. 4 are thus the primary candidates for observing the elusive CDW gap in VSe$_2$.

## 5   Discussion and conclusion

We have shown that the structured susceptibility in $1T$-VSe$_2$, including the momentum-dependence of the electron-phonon coupling, shows a sharp peak at the experimentally observed CDW ordering vector. Moreover, its temperature dependence reproduces the thermal evolution of the CDW wave-vectors observed by X-ray diffraction experiments. Our results demonstrate that this thermal variation is an intrinsic effect, whose observation does not necessitate a description in terms of multiple consecutive CDW phases. Additionally considering the error margins of the reported X-ray diffraction data (see Fig. 3), the lack of indicators for a second transition in thermodynamic probes [15,22,23], and the fact that satellite dark-field phase contrast may be due to a natural phase variation of the CDWs [34], we suggest that VSe$_2$ hosts a single CDW phase. The resolution of this discussion by the effect of a structured electron-phonon coupling brings VSe$_2$ in line with more strongly coupled CDW materials in which temperature-dependent incommensurate CDW ordering wave-vectors are commonly observed [44].

Based on a computation of the spectral function, we predict the elusive CDW gap in VSe$_2$ to appear as localized suppressions of spectral weight centered above $E_F$, most pronounced on the sides of the Fermi surface lobes around $k_z = 0.17c^*$ and $k_z = 0.5c^*$. The degree of localization and offset from the Fermi energy reflect the weakness of the nesting in this system. High-resolution, $k_z$-resolved ARPES at varying temperature should be able to resolve the opening of the predicted CDW gaps.

The results reported here for the specific CDW material VSe$_2$ fit into a larger picture of structured electron-phonon coupling being essential to the quantitative understanding of any charge ordered material. That is, in an ideal single-band, one-dimensional (1D) model for a metal, a Peierls transition may be signalled in the Lindhard function, which describes the (bare) electronic susceptibility and which diverges in 1D metals at the wave-vector $Q = 2k_F$ connecting the two Fermi surface points [24,50]. In real materials, however, perfect FS nesting never occurs, and inspection of the Lindhard function commonly indicates either no clear peak, or a dominant peak at a wave-vector inconsistent with the observed CDW [51]. Ex-

amples of supposedly nesting-driven density wave materials for which this was demonstrated explicitly include $2H$-NbSe$_2$ and $2H$-TaSe$_2$ [19, 20, 51, 52], TbTe$_3$ and other rare-earth tellurides [51, 53–55], blue bronze (K$_{0.3}$MoO$_3$) [56][2], and chromium [57]. Even for materials with electronic band structures that are considered well-nested, the assumption that density waves arise purely from FS nesting is thus demonstrably incomplete.

In contrast, the structured susceptibility, which includes the momentum and orbital dependent electron-phonon coupling, has been shown to agree with experimental observations in a range of real CDW materials. For example, in the prototypical strong-coupling, quasi-two-dimensional CDW compound $2H$-NbSe$_2$ incorporating the momentum and orbital-dependence of the EPC was shown to correctly predict the wave-vector of its electronic instability [19, 20, 43, 58]. Similarly, the concept of "hidden nesting", taking into account the real-space shape and orientation of valence orbitals in CDW formation, effectively corresponds to including an orbital-dependent EPC [54, 59–61]. The need for including a coupling structure more generally, however, is typically associated with the strong-coupling nature of specific materials, and is far from standard practice [24, 51, 53, 55, 56, 62].

The results presented here for the weakly coupled, single-band material VSe$_2$ highlight the need for considering the structure of the electron-phonon coupling in quantitative models for any density wave material, regardless of its dimensionality, coupling strength or degree of nesting. Besides the momentum-dependence considered here, the coupling may in general also depend on orbital character and even spin. All of these contribute to the physical properties of density wave materials, and understanding their quantitative impact is indispensable in understanding the emergence of and interaction between charge, orbital, and magnetic order throughout (unconventional) superconductors, magnets, and (multi)ferroics.

# Acknowledgements

The authors would like to acknowledge Yang-Hao Chan for sharing extra details of the phonon dispersion computation presented in [41]. F. F. acknowledges support from the Astor Junior Research Fellowship of New College, Oxford.

**Author contributions** All authors contributed to the planning, research, and writing involved in this work.

# A  Supplementary figures

## A.1  Degree of nesting

To quantify the degree of nesting in VSe$_2$, one can consider the so-called nesting function $\lim_{\omega \to 0} \mathrm{Im}[\chi]/\omega$, which is expected to diverge at vectors $\mathbf{q}$ that nest the Fermi surface [51]. Numerically, a finite value for $\omega$ is required; considering $\omega = 1$ meV corresponds to computing the degree of "nesting" between states around the Fermi level separated in energy by 1 meV. In Fig. 5, we plot $\mathrm{Im}[\chi]/\omega$, with $\chi(\mathbf{q}, \omega)$ the Lindhard function. The absence of pronounced peaks in these plots, and the fact that the maximum in Fig. 5 does not coincide with the experimentally observed CDW propagation vector, show that VSe$_2$ is not a well-nested material.

---

[2]In their Sec. IV C, they note that they added a shift to their Lindhard function to make it fit to experimental data, which exhibits a temperature dependence their calculations cannot account for.

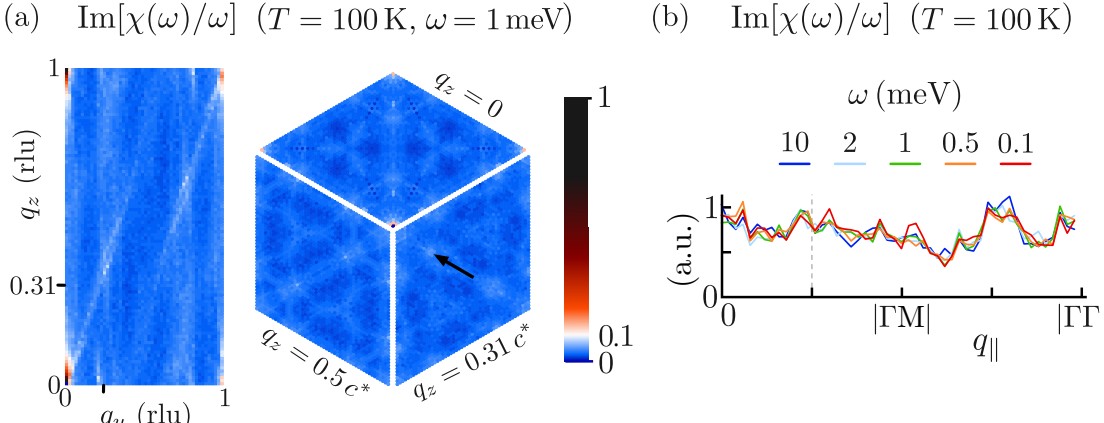

Figure 5: (a) The magnitude of the nesting function $\text{Im}[\chi]/\omega$, with $\chi(\mathbf{q}, \omega)$ the Lindhard function and $\omega = 1\,\text{meV}$. The rectangular figure is in the $q_y q_z$-plane, with $q_x = 0$, while the hexagonal figure shows the function in the $q_x q_y$-plane for three different values of $q_z$. The ticks on the axes of the rectangular figures indicate the location of the observed CDW ordering wave-vector, also indicated by black arrows in the hexagonal figures. Note the lack of a significant peak at this position. (b) The nesting function for $\mathbf{q}$ along the black arrow shown in (a), for different values of $\omega$. There is no clear nesting peak.

## A.2 Further analysis of $\chi$ and $D_2$

Fig. 6 compares the variation with momentum of the structured susceptibility $D_2$ and Lindhard function $\chi$, at various energies. Although at $E_F$, the out-of-plane component of the peak position of $D_2$ was shown in Fig. 3 of the main text to closely track the experimentally observed values, the highest peak in the susceptibility actually lies at $q_y = 0.26\,\text{rlu}$ at that energy[3]. This may be explained by the fact that we neglect any lock-in effects in the present computation, which would favor an adjustment of the in-plane component towards the commensurate value of 0.25 rlu. Comparing $D_2$ and $\chi$, it is evident that the structure of the electron-phonon coupling selectively amplifies and suppresses the various sub-peaks that make up the Lindhard function. These sub-peaks arise from similar wave-vectors connecting pairs of states in different regions of the Fermi surface, so that they generically have different associated electron-phonon coupling strengths.

Fig. 7 shows the influence of small chemical potential shifts on the temperature-dependence of the maxima in the Lindhard function $\chi$ and the structured susceptibility $D_2$ along the line $\mathbf{q} = (0, 0.25, q_z)$. Variations in the chemical potential can for instance arise from the non-stoichiometry of crystal samples, and might lead to small variations in the experimentally determined CDW wave-vectors. Regardless of the exact chemical potential shift applied, the structured susceptibility shows a clear trend towards a smooth (downward) variation of the charge-ordering wave-vector as the temperature is decreased. The maximum of the much flatter function $\chi$ does not show any such trend. This figure demonstrates that the structure of the electron-phonon coupling can significantly affect not only the position of the maximum of the susceptibility, but also its temperature-dependence.

---

[3]Notice that that we are limited by the resolution of the band-structure calculation, $\delta q_x = \delta q_y = 0.01$ rlu, and $\delta q_z = 0.0025$ rlu.

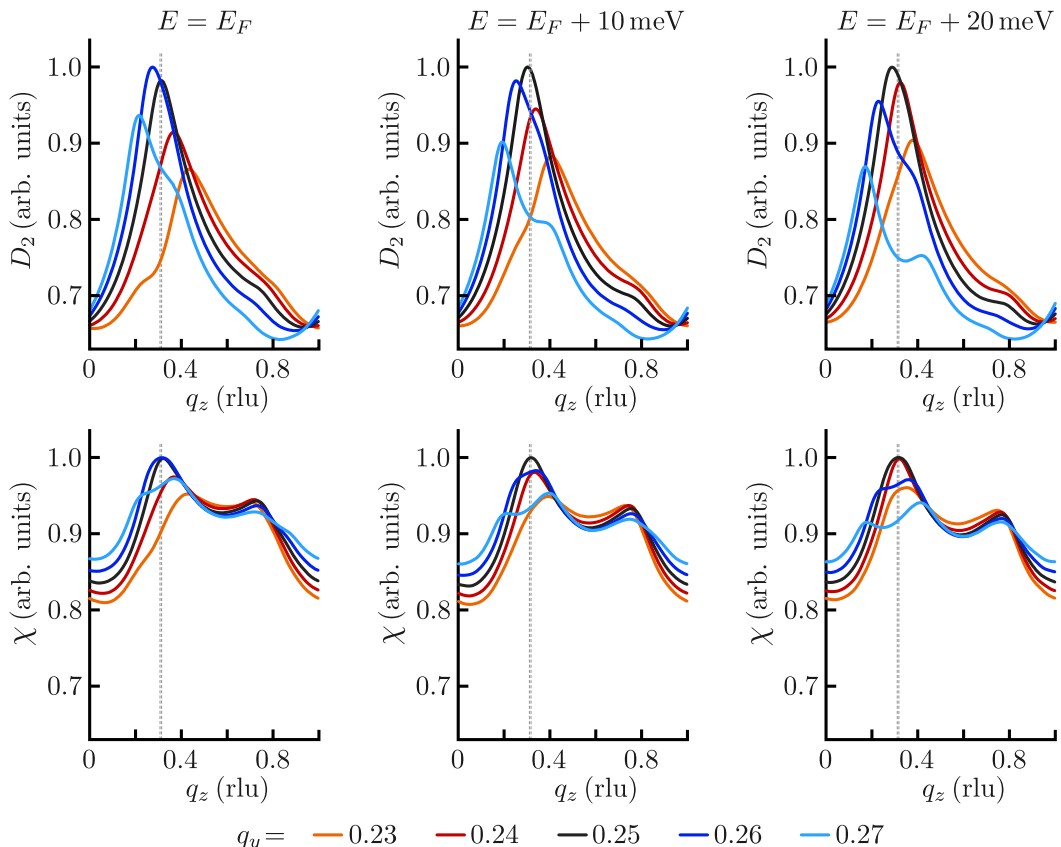

Figure 6: The structured susceptibility (top row) and Lindhard function (bottom row) along $q_z$, for various values of $q_y$, and varying the chemical potential. The two closely-spaced vertical dashed lines correspond to the two experimentally observed values of the out-of-plane component of the CDW wave vector [21]. Curves are normalised per panel.

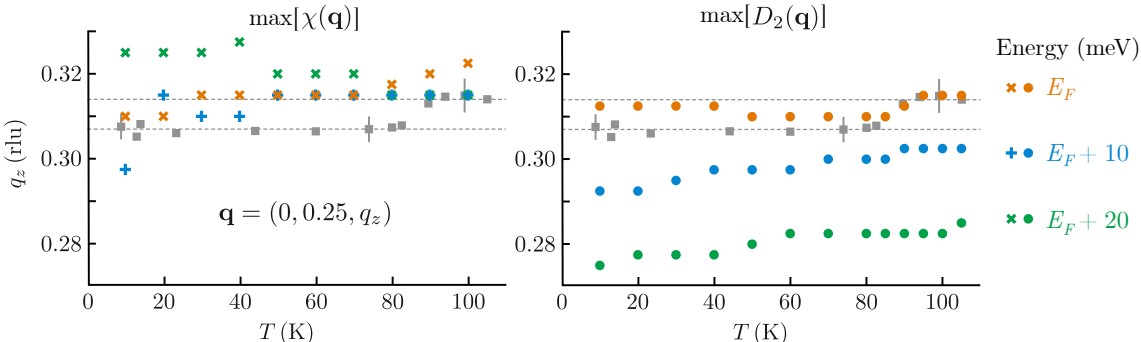

Figure 7: The temperature dependence of the maxima of the Lindhard function $\chi$ and structured susceptibility $D_2$ along the line $\mathbf{q} = (0, 0.25, q_z)$, varying the chemical potential. $E_F$ is the experimentally determined Fermi level, which lies 20 meV below the zero energy level of the computed *ab initio* band structure shown in Fig. 1 of the main text. The grey data points indicate the experimentally determined charge ordering wave-vectors from [21], and the grey dashed lines indicate $q_z = 0.307c^*$ and $q_z = 0.314c^*$. While $D_2$ shows a clear trend in its temperature dependence, $\chi$ does not.

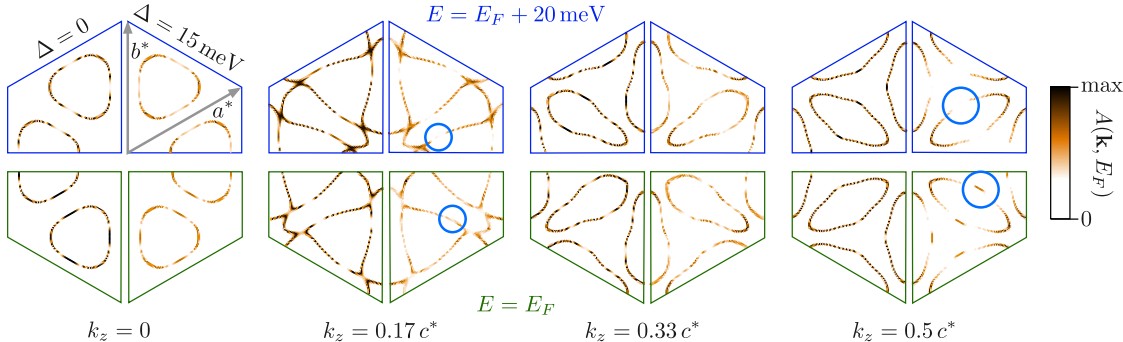

Figure 8: The spectral function of $VSe_2$, for various values of $k_z$. The left(right) side of each plot shows the ungapped(gapped) phase above(below) $T_c$. The top(bottom) half of each plot shows the spectral function 20 meV above $E_F$ (at $E_F$). All plots employ the same colour scale. We employed a spectral broadening of $\Sigma = 15i$ meV, and a constant gap $\Delta = 15$ meV. The blue circles highlight regions regions separated by one CDW wave-vector.

## A.3   Spectral function at different energies

Turning to the spectral function in Fig. 8, we contrast the spectral function of $VSe_2$ in the presence of a CDW gap with the situation at $\Delta = 0$. The top half of each hexagon shows the spectral function 20 meV above the Fermi level, and reproduces Fig. 4 of the main text. The bottom halves contrast this with the spectral functions at the Fermi level. The absence of a gap at $E_F$ is highlighted by the blue circles.

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
