# Peer review of "Charge order from structured coupling in VSe$_2$"

_SciPost Physics, doi:SciPost Phys. 9, 056 (2020)_

## Round 2 · Referee Report · Anonymous · 2020-9-1

Strengths

1- Very high level of technical execution
2- Sound conclusions on a well-known and challenging problem
3- Suggest further experiments, which will certainly stimulate future research
4- The manuscript is well-organised and clearly written

Report

This work addresses the charge-density-wave (CDW) phases of bulk 1T-VSe2 using state-of-the-art theoretical approaches. This materials is rather different from closely related dichalcogenides of Nb and Ta where the CDW phases are understood better. The important conclusion of this work is that the temperature dependence of q_z is likely a result of one single CDW phase, and not two competing phases as often assumed. Overall, the work present a very complete picture CDW physics in 1T-VSe2.

This is very high quality work arrives to new and original conclusions related to a very challenging problem. I recommend publishing it as is.

---

## Round 2 · Referee Report · Anonymous · 2020-9-13

Strengths

1- This work highlights the need for considering the structure of the electron-phonon coupling in quantitative models for density wave in VSe2.
2- Quantitative agreement with experimental measurements.

Report

In this paper, the authors study theoretically the charge order in VSe2 compound. From DFT electronic bands, they calculate the electronic susceptibility, the charge density wave (CDW) vector and CWD gap. Using a numerical method, that some of the co-authors had already used for other systems recently, they can evaluate the “structured susceptibility” taking into account the momentum-dependent electron-phonon coupling. This method allows them to report quantitatively on experimental results that are not reproduced correctly by the usual methods of calculation. The authors thus show the importance of the structured electron-phonon coupling to get a correct theoretical description of the charge density waves in VSe2. This work is motivated by many unexplained experiments in a very active field. It brings a major breakthrough in the understanding of CDW for this type of system. This manuscript is well presented and written in a very clear way. Therefore, I consider that the manuscript is suitable for publication in SciPost.

Requested changes

I only have three minor questions:

(1) In the caption of figure 1, the authors write “Orange corresponds to Se character, and blue indicates V d-orbital character.” The p-orbital character of Se is not written, does it mean that Se character of states around EF is not only p-character ?

(2) In the calculation of the spectral function, a spectral breadening of 10 meV is used (caption of figure 4). Is not this value too close to the gap of 15 meV to give good results (ie results independent of the breadening value) ?

(3) Figure 4 and figure 8: the maximum value of the spectral function is not given. Is this maximum the same for all panels or does it change depending on the panel?

  • validity: high
  • significance: high
  • originality: high
  • clarity: high
  • formatting: excellent
  • grammar: -

Author:  Jans Henke  on 2020-09-21  [id 979]

(in reply to Report 2 on 2020-09-13)
Category:
answer to question

Dear referee,
We would like to thank you for your time and effort in reviewing our submission, and for your positive comments regarding our work. We have made minor revisions in response to the three minor questions posed in your referee report, please see our response below:

Q1: In the caption of figure 1, the authors write “Orange corresponds to Se character, and blue indicates V d-orbital character.” The p-orbital character of Se is not written, does it mean that Se character of states around EF is not only p-character?

A1: We thank you for pointing out this source of confusion. The colours in Figure 1a do not resolve orbital character explicitly, so we cannot say orange is purely p-character, nor that blue is purely d-character. In spite of this, we know the orange bands must be mostly p-character, and the blue bands d-character, given the energies and grouping of the bands. To remove the source of confusion, we have removed "d-orbital" from the figure caption.

Q2: In the calculation of the spectral function, a spectral broadening of 10 meV is used (caption of figure 4). Is not this value too close to the gap of 15 meV to give good results (ie results independent of the broadening value) ?

A2: We fully agree that care should be taken with the amount of spectral broadening compared to the size of the gap. There are several factors that play a role here. First, in the limit of a very large spectral broadening, one expects to not be able to resolve any small gaps in the spectral function, because any gaps will be smeared out entirely. Second, the smaller the spectral broadening, the denser the k-mesh that one needs to compute the spectral function on. When the k-mesh is not dense enough, this leads to artefacts such as the oscillations seen along the spectral contours in the ungapped plots in Figs. 4 & 8. This sets an effective lower limit on the size of the spectral broadening. In the attachment we demonstrate what is obtained for different values of the spectral broadening (in the same colourmap as Figs. 4 & 8, each of the three sets of four panels normalised to their respective maximum), where this effect is apparent. Lastly, for comparison to experiment it is more meaningful to use an experimentally realistic spectral broadening. 15 meV is a reasonable broadening to use compared to the energy resolution of current ARPES equipment in combination with broadening effects from impurity scattering, finite domain-size, the effects of rotational disorder and imperfect cleavage surfaces.

Given that with a spectral broadening of 15 meV we can still see the localised spectral weight suppression with the k-mesh that we used, we conclude that the gaps we identify in Figs. 4 & 8 are robust and should be detectable with existing ARPES equipment.

Q3: Figure 4 and figure 8: the maximum value of the spectral function is not given. Is this maximum the same for all panels or does it change depending on the panel?

A3: All panels are shown in the same colour scale. This is an important point and we agree with the referee that it should have been made clearer. We have added the sentence “All plots employ the same colour scale.” to the figure captions of Figs. 4 and 8.

Attachment:

---

## Round 3 · Referee Report · Anonymous (Referee 2) · 2020-10-6

Report

In their response the authors addressed my comments and slightly modified the manuscript accordingly. The detailed answer to the technical question Q2 allows a better understanding and justification of the relevance of the presented results. This confirms the importance of this work for a better understanding of experiments (in particular ARPES measurements).
In line with my previous report I therefore recommend this manuscript for publication in SciPost Physics.

---

## Round 3 · Author Response

We would like to thank the referees and editorial team for their time and effort in reviewing our submission, and for their many positive comments regarding our work. We have made minor revisions in response to the three minor questions posed in Referee Report 2 - for the response to these questions, please see our reply to this report.

---

## Round 3 · List of Changes

• Removed "d-orbital" from the caption of Figure 1.
  • Added the sentence "All plots employ the same colour scale." to the captions of Figures 4 & 8.
  • Fixed a typo in Figure caption 4: the spectral broadening is 15i meV rather than 10i meV.

---

## Editorial Decision

published